# Token Representation Shrinkage
# Impairs Creativity of Generative Models

## Abstract

Transformer-based generative models have been widely used for generating high-quality images and other continuous data modalities. Despite their widespread adoption, these models frequently exhibit limitations in creativity, often failing to produce diverse and novel outputs. Most existing studies analysing these shortcomings have predominantly concentrated on enhancing the generative architecture or training methodologies. In contrast, our study shifts the focus to the tokenization process, exploring how discretizing continuous representations into discrete tokens influences the overall creativity of generative models. Through systematic analysis, we identify a critical phenomenon we term "token representation shrinkage," characterized by the collapse of representation diversity within discrete codebook tokens and their continuous latent embeddings in vector quantization, which is one of the most popular discrete tokenization method used. Our findings reveal that this shrinkage problem significantly reduces the creativity of generative models, adversely affecting performance across various domains, including natural images and real-world medical images.

## 1 Introduction

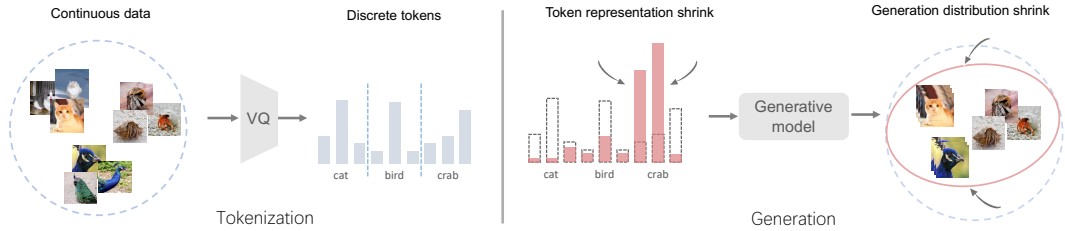

Figure 1: **Token representation shrinkage leads to diversity loss in generative model.** Left: Vector quantization is a widely used technique to map continuous data into discrete token which enable the generative model's generation. Right: We observe that token representation shrinkage, manifested as narrow distribution in latent space, leads to a shrunk distribution of the generated data.

Transformer-based generative models for autoregressive generation have gained significant popularity in recent years in the field of image generation. These models underpin many state-of-the-art systems such as DALL-E [3] and VAR [25], which have found wide-ranging applications in art creation, design automation, and data augmentation. Their practical value lies not only in producing visually compelling images but also in enabling new workflows for creative and industrial domains.

Despite their success, transformer-based generative models suffer from a widely observed issue: the synthetic images they generate often exhibit a narrower distribution compared to original images. This phenomenon, commonly referred to as mode collapse, results in limited diversity in the generated content. Mode collapse leads to a loss of diversity in generated outputs, causing the model to ignore valid variations in the data distribution, which limits its generalization, realism, and utility in downstream tasks. In this study, we refer the ability of generative models to produce diverse high quality outputs as creativity. Therefore, mode collapse and limited diversity in output will lead to decreased creativity of the generative models.

Most existing studies on these problems have predominantly focused on the generative architecture or training methodologies. To address this, various studies have proposed architectural innovations or alternative training objectives. For example, VQGAN [10] incorporates vector quantization to learn a diverse discrete codebook, while ImageGPT [5] treats images as sequences of pixels to better capture complex data distributions and enhance generative diversity.

However, in this work, we identify a previously overlooked but critical factor in tokenization, termed token representation shrinkage, which contributes to the decline in generative creativity. Specifically, the root of this problem lies in a core component of transformer-based image generators: the use of vector quantization (VQ), one of the most widely used discrete tokenizers, for tokenizing images. VQ is crucial for converting continuous image features into discrete tokens suitable for transformer processing. However, we find that when the token representation distribution undergoes shrinkage, the generative model's output creativity is significantly reduced. As shown in Fig. 1, VQ techniques map continuous data into discrete tokens. However, when tokens shrank into a limited region of the distribution, the generated outputs are also constrained to a narrow portion of the data space, resulting in reduced diversity and diminished modality coverage.

We further identify a specific mechanism that contributes to token representation shrinkage: the commonly used token initialization strategy during VQ training. Typically, token embeddings are initialized based on the outputs of an untrained encoder, which results in a clustered initial token distribution. This initialization bias suppresses the token space's ability to expand during training, preventing it from aligning with the true data distribution and thus inducing representation shrinkage.

To address this, we propose a simple yet effective solution: pretrain the encoder without VQ and then fine-tune it with VQ enabled. This approach allows the encoder to learn meaningful semantic representations before quantization is introduced, thereby reducing the resistance faced during VQ optimization and alleviating the token shrinkage effect. We validate our hypothesis and proposed method through extensive experiments on both synthetic datasets and real-world datasets, including ImageNet, CIFAR-10, and the Ocular Disease Recognition medical dataset. Our results demonstrate that token representation shrinkage leads to decreased generative creativity and that our approach significantly mitigates this issue, improving both diversity and fidelity of generated images.

Our main contributions are summarized as follows:

- We identify a previously underexplored cause of mode collapse in transformer-based generative models: token representation shrinkage.

- We provide a detailed analysis of how poor token initialization contributes to this phenomenon.

- We propose a simple and effective training strategy, pretraining without VQ followed by fine-tuning with VQ, to resolve the issue.

- We empirically validate our findings on both synthetic and real-world datasets, demonstrating improved generative performance.

## 2   Related Works

Vector Quantization is foundational in data compression and signal processing per Shannon's rate-distortion theory [12, 7] , traditionally relied on methods like K-means clustering [19] but faced high complexity with high-dimensional data [17]. To mitigate this challenge, DeepVQ [17] improved efficiency by mapping data to lower-dimensional latent spaces before quantization. Moreover, [26] proposed VQ-VAE which integrates VQ with variational autoencoders, using a straight-through estimator [2] to handle discrete variables. To refine VQ methods for improved performance, variants

such as Residual Quantization [18], Product Quantization [6], and Soft Convex Quantization [11] further enhanced representation capacity and efficiency. Recent advances incorporate attention mechanisms and transformer architectures [27, 28] to dynamically select codebooks and capture global data dependencies. Recent works also explore per-channel codebooks [14] and neural network variants of residual quantization [15] to predict specialized codebooks, enhancing the model's expressive power.

VQ has been widely applied across various domains. In natural language processing, VQ facilitates sequence modeling [16] enhancing tasks such as language modeling. In computer vision, VQ has significantly advanced image generation and compression techniques [10]. Similarly, in audio processing, VQ techniques have captured complex temporal dependencies [8]. Furthermore, in multimodal applications, VQ supports the integration of different data types through shared discrete representations [23].

Despite these advancements, VQ methods encounter challenges that restrict their broader application, including but not limited to codebook collapse, training instability, and computational overhead. Extensive research has been conducted on solving the codebook collapse problem, where only a subset of tokens are used leading to inefficient representation usage and reduced diversity in outputs, by reducing token dimension [28], orthogonal regularization loss [24], multi-headed VQ [20], finite scalar quantization [22], and Lookup Free Quantization [29]. Recent methods like [13] and [1] also strive to enhance tokens usage efficiency. However, beyond the widely recognized issue of codebook collapse, our work identifies, investigates, and proposes potential solutions for collapses of tokens and reconstruction, which pose serious challenges to VQ and merit attention.

# 3 Preliminary

## 3.1 Definition of Creativity for Generative Model

In this study, we define the *creativity* of a generative model as the diversity of high-quality content it generates. For example, an ideal image generative model should produce high-fidelity images which are very different from each other. Most previous works related to creativity of generative models focus their research on the generative models [10, 5]. However, we observe that shrinkage of token representation distribution is also an important factor to consider for creativity. Our experiments suggest that **token representation shrinkage** significantly impairs the creativity of transformer-based generative models.

## 3.2 Preliminary of Vector Quatization

**VQ-VAE**   We define the VQ-VAE as following: an encoder $E_\theta$, a decoder $D_\theta$ and a set of tokens $\mathcal{T} = \{t_1, t_2, \ldots, t_S\}$. The token set $\mathcal{T}$ constitutes the codebook, which is employed to store the discretized representations. The encoder is responsible for mapping the raw data $X = \{x_1, x_2, \ldots, x_N\}$ to a set of continuous representations $\mathcal{Z} = E_\theta(X)$, where $\mathcal{Z} = \{z_1, z_2, \ldots, z_N\}$. And the decoder reconstructs the data $X' = D_\theta(\hat{Z})$ based on the set of discretized representations $\hat{Z}$, where $\hat{Z} = \{\hat{z}_1, \hat{z}_2, \ldots, \hat{z}_N\}$. The process of tokenizing a continuous representation $z_j$ to discrete representation $\hat{z}_j$ is as following:

$$\hat{z}_j = \arg \min_{t_k \in \mathcal{T}} \|z_j - t_k\|, \tag{1}$$

where $t_k$ is a token in token set $\mathcal{T}$ and $k$ is the index. This quantization is performed by finding the nearest token $t_k$ in $\mathcal{T}$. The optimization objective comprises reconstruction loss $\mathcal{L}_{\text{recon}}$, codebook loss $\mathcal{L}_{\text{codebook}}$, and commitment loss $\mathcal{L}_{\text{commit}}$. Additionally, we adopt the exponential moving averages (EMA) adopted by [26] to update the codebook instead of the codebook loss term.

**Initialization Strategy**   For codebook initialization, a widely used initialization strategy is K-means[30]. It uses the encoder output $\mathcal{Z}$ and perform K-means algorithm to initialize the tokens $\mathcal{T}$, where $N$ is the number of encoder output and $S$ is the number of tokens. The initialization aims to minimize the total distance from each vector $z_j$ to its nearest token $t_k$. The optimizing function is shown in equation 2,

$$\min \sum_{j=1}^{N} \sum_{k=1}^{S} r_{jk} \|z_j - t_k\|^2, \tag{2}$$

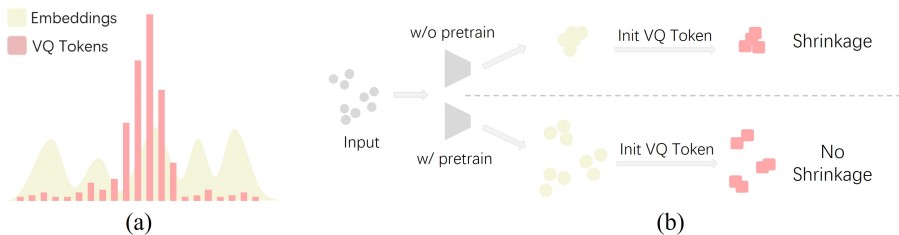

Figure 2: **Token representation shrinkage phenomena is attributed to biased initialization.** (a) Token representation shrinkage refers to the phenomenon where token becomes concentrated on a small number of modes, despite the original continuous embeddings exhibiting diverse and well-separated modes. (b) Our analysis suggests that token representation shrinkage arises from initializing tokens with untrained embeddings that lack sufficient modality information.

where $r_{jk} = 1$ if $z_j$ is assigned to cluster center $t_k$, otherwise $r_{jk} = 0$ .

## 4 Token Representation Shrinkage Problems

This section presents an analysis of the token representation shrinkage phenomenon and investigates its underlying causes.

### 4.1 Shrinkage Phenomena and Sythentic Experiments Results

Token representation shrinkage is characterized by a disproportionate concentration of tokens around a limited subset of encoder output embeddings, as shown in Fig. 2 (a). This shrinkage results in a poor representation since the ideal scenario requires a fitting distribution of tokens that effectively aligns with the underlying embedding space.

To validate the token representation shrinkage phenomenon, we conduct experiments on our synthetic dataset using VQ-VAE. Specifically, we use VQ-VAE to reconstruct the input data and compare the resulting token distribution with the original data distribution. The synthetic dataset comprises 10,000 data points, uniformly sampled from 10 distinct Gaussian distributions (see Sec. 5.1 for details). As shown in Fig. 3 (a), (c), and (e), tokens densely cluster within a specific region of the latent space, which subsequently causes the reconstructed data to collapse. As a result, the reconstructions fail to capture the full modality spectrum of the original data.

One contributing factor to token representation shrinkage is the clustering of token embeddings during codebook initialization. This occurs when the initial embeddings are distributed within a narrow region of the latent space, limiting their expressiveness and leading to early-stage shrinkage. As shown in Fig.2 (b), the output distribution of an untrained encoder is significantly more concentrated compared to that of a trained encoder.

In order to examine how untrained encoder initialization contributes to token representation shrinkage, we compare the embedding distributions produced by trained and untrained encoders on the synthetic dataset. We observe that the untrained encoder produces embeddings that are concentrated in a narrower region and exhibit fewer distinct peaks, suggesting that they represent fewer, less distinguishable modes. This supports the conclusion that token representation shrinkage is primarily caused by the use of untrained encoders for token initialization. Since the untrained encoder lacks the capacity to extract meaningful features from the input data, it maps diverse inputs to similar embeddings, leading to a poorly distributed token initialization and reduced representational diversity. Further experimental details and visualizations are provided in the supplementary material.

Building on these observations, we hypothesize that if tokens are initialized based on encoder that has learned semantic distinctions and its output embeddings are dispersed, it would enhance the semantic distinction among tokens and thus control token representation shrinkage. Consequently, we propose a straightforward yet effective method to mitigate token representation shrinkage: pretrain without VQ, then fine-tune with VQ. It first trains an autoencoder, and then trains the VQ-VAE initialized with the weights of the autoencoder trained at the first stage. Pretraining the encoder allows it to

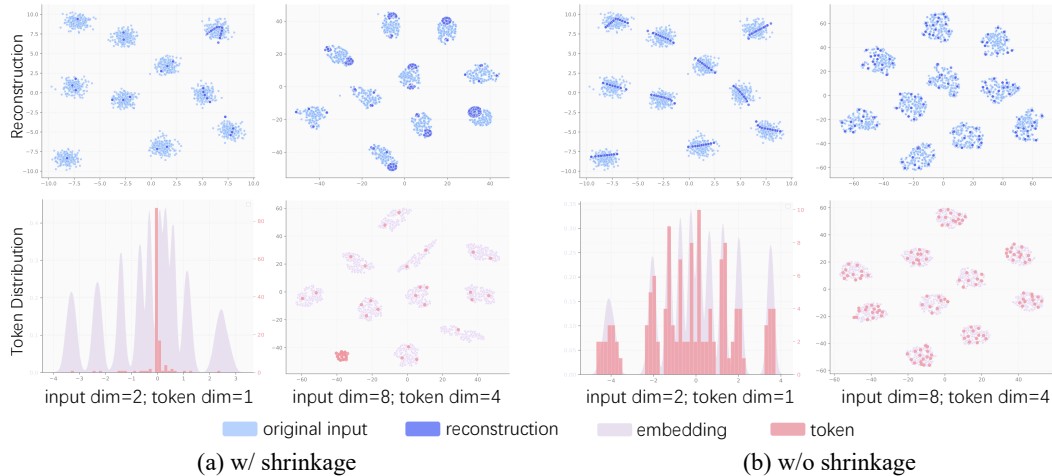

(a) w/ shrinkage            (b) w/o shrinkage

Figure 3: **Visualization of token shrinkage effects in synthetic experiments**. (a) With shrinkage: most of the token distribution clusters into a narrow region of the embedding space, leading to a loss of diversity across reconstruction modes under different input dimensions. (b) Without shrinkage: tokens are well distributed across the embedding space, enabling accurate and diverse reconstructions.

discern differences in input data, resulting in more distinctly spaced embeddings, providing a robust foundation for initializing the tokens, as demonstrated in Fig. 2 (b).

We evaluate the effect of our pretraining strategy on the synthetic dataset, with results shown in Fig. 3. The comparison between subfigures (a) and (b) demonstrates that when the shrinkage problem is mitigated by our pretraining approach (Fig. 3 (b)), the resulting token distribution becomes more uniform, and the reconstruction aligns more closely with the original input distribution. Notably, under higher input dimensions (input dim=8), the token shrinkage problem still leads to degraded reconstruction performance. In contrast, the version without shrinkage produces a reconstruction distribution that aligns more closely with the original data distribution. This suggests that addressing token shrinkage is critical for enhancing the creativity (diversity and quality) of generative models.

## 4.2 Formal Definition of Token Representation Shrinkage

To mathematically analyze the token representation shrinkage effect, we consider a data distribution constructed from $K$ well-separated and equally weighted component distributions $p(x|k)$,

$$p(x) = \sum_{k=1}^{K} p(x|k)p(k) = \frac{1}{K} \sum_{k=1}^{K} p(x|k), \tag{3}$$

where $p(k) = \frac{1}{K}$ because of equal weights. For simplicity, we assume that both the encoder and decoder are identity mapping (i.e. $X' = Dec(Enc(X)) = X$), and that the transformer can perfectly model the full token distribution. Under these assumptions, the only source of distortion arises from *vector quantization*. Accordingly, the expected mean squared error in pixel space roughly express the upper bound of generation quality:

$$\mathcal{E} = \mathbb{E}_{x \sim p} \left[ \|q(x) - x\|_2^2 \right], \tag{4}$$

where $q$ is quantization function. We assume the entropy of the generated mode distribution measure the diversity:

$$H = - \sum_{k=1}^{K} p_k \log p_k, \quad \text{where} \quad p_k = \frac{|T_k|}{\sum_{j=1}^{K} |T_j|}, \quad T_k = \{t_i \mid t_i \in \text{cluster } k\}, \tag{5}$$

where $T_k$ is the set of tokens assigned to cluster $k$, and $p_k$ is the empirical probability (proportion) of tokens in that cluster.

In the ideal case of balanced token utilization, we expect $|T_k| \approx S/K$, yielding $p_k \approx 1/K$, maximal entropy $H = \log K$, and minimal quantization error $Q$. However, under **token representation shrinkage**, token becomes concentrated in a subset of modes $\mathcal{J} \subset \{1, \ldots, K\}$, where $|\mathcal{J}| = M \ll K$. This leads to a reduced entropy

$$\Delta \mathcal{E} = logM - logK < 0, \tag{6}$$

which means the diversity will be impaired. Moreover, samples from inactive modes $k \notin \mathcal{J}$ are forced to encode using distant tokens, thereby increasing the quantization error and subsequently decreasing the generation quality.

# 5 Experiments Design and Results

In this section, we firstly conduct experiments on CIFAR-10 to validate the existence of token representation shrinkage in the real-world dataset. And then we demonstrate that token representation shrinkage negatively impacts the creativity of generative models, thereby decreasing both the diversity and fidelity of generated samples.we conduct experiments on two representative generative models, MaskGIT [4] and VAR [25], using both the ImageNet-100 dataset and a medical image dataset.

It is important to note that in the experiments involving generative models, the use of GAN-based losses can introduce smoothing effects to the model, potentially hallucinating the presence of token representation shrinkage. Therefore, in this section, we adopt VQ-VAE as the image tokenizer. The training loss includes *codebook loss*, *commitment loss*, *MSE loss*, and *perceptual loss*. Generative experimental results based on VQGAN are available in the supplementary.

## 5.1 Experiment Setup

**Dataset**    As mention in in Sec. 4.1, we conduct experiments on a synthetic dataset to validate our hypothesis regarding the causes of token representation shrinkage. The synthetic dataset consists of 10,000 data points, obtained by sampling 1,000 points from each of 10 Gaussian distributions with identical standard deviations but distinct means. This setup yields ten equally sized classes with similar distribution, designed to emphasize disproportionate token allocation and make token representation shrinkage patterns more easily observable. To investigate token representation shrinkage behavior under varying data complexity, we generate synthetic datasets with different input dimensionalities. And to further validate existence of token representation shrinkage, we adopt CIFAR-10 to do conduct experiments.

For experiments regarding generative model, we adopt ImageNet-100 which is a subset of the ImageNet-1K dataset containing 100 classes. The original ImageNet-100 comprises approximately 130,000 training images and 5,000 test images. To better evaluate both reconstruction-FID (r-FID) and generation-FID (g-FID), we uniformly sampled total 20,000 images from all training classes to build up test dataset and construct an additional validation set containing 5,000 images. For the medical domain, we adopt the Ocular Disease Recognition (ODIR) [21] dataset, which contains 6,716 fundus images labeled across 8 diagnostic categories. We using a 70%/20%/10% split to partition the data into training, test, and validation sets.

**Metrics**    For the synthetic dataset, we directly visualize the original data and its reconstructions, along with the corresponding token and embedding distributions, as shown in Fig. 3. For high-dimensional data, t-SNE is applied for dimensionality reduction prior to visualization.

To quantify the token representation shrinkage problem, we utilize cosine distance and perplexity. The average pairwise cosine distance across the codebook serves as an indicator of code clustering, with lower values suggesting that the code vectors have concentrated in a limited angular region. The perplexity, which is computed by the entropy over the codebook likelihood, reflects the effective tokens being utilized and is maximized when all tokens are used uniformly.

To evaluate the tokenizer's reconstruction performance, we adopt reconstruction FID (r-FID), mean squared error (MSE), and LPIPS scores. For generative quality, we utilize generation FID (g-FID) as the primary metric. To assess the diversity and distributional coverage of generated samples, we compute the average pairwise pixel-level distance between generated images.

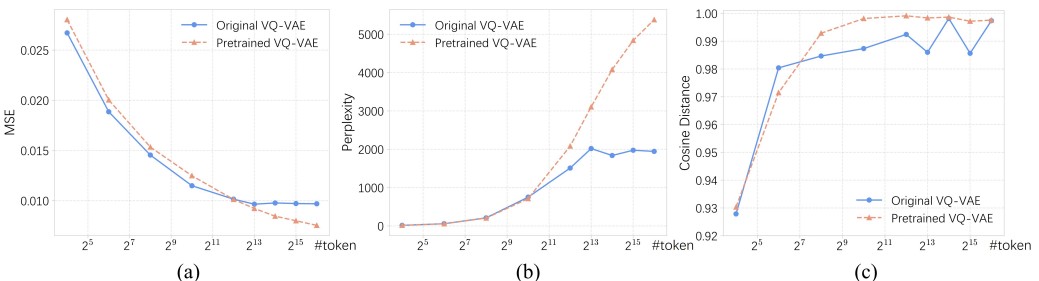

Figure 4: **Validation of token representation shrinkage on CIFAR-10** (a) Token representation shrinkage lead high reconstruction errors. (b) Token representation shrinkage leads to lower perplexity, demonstrating lower token utilization. (c) Token representation shrinkage leads to higher similarity among token during training.

**Training Configuration** For generative model experiments, we follow the tokenizer framework proposed in VQGAN [10]. Due to resource limitation, we resize all input images to $128\times128$ resolution and reduce the backbone's channel size to 64. To preserve a $16\times16$ latent spatial resolution, one downsampling layer and its upsampling layer are removed. All tokenizer experiments are conducted with a fixed codebook size of 16,384. To ensure feasibility under limited resources, we use the smallest generative model configurations. The MaskGIT generator employs a ViT [9] with depth of 24, while the VAR model uses a depth of 16. All tokenizers and generative models are trained on 2 A100 GPUs with 40 GB memory. Training the tokenizers on ImageNet-100 typically takes 1.5 to 3 days, while training the generative models requires 3-6 days depends on setting. Complete training details and hyperparameters are provided in the supplementary material.

## 5.2 CIFAR-10 Results

To validate that the shrinkage exists under real-world data conditions, we conducted corresponding experiments on the CIFAR-10 dataset. Additionally, we hypothesize that given a fixed dimensionality of the representation space, an increase in the number of tokens tends to facilitate their clustering, thereby making token representation shrinkage more pronounced. Under these conditions, the disadvantages caused by token representation shrinkage problem likely become more evident. Therefore, we evaluated the performance VQVAE's performance across varying token quantities.

As shown in Fig. 4, the original VQ model performs well when the number of tokens is relatively small. However, as the token count increases, particularly beyond $2^{12}$, its reconstruction performance deteriorates relative to the pretrained counterpart. Notably, the perplexity curve of the original VQ flattens after $2^{13}$ tokens, indicating poor token utilization. Additionally, its average cosine distance remains consistently lower than that of the pretrained model, suggesting a higher degree of similarity among tokens. These findings collectively indicate that the token shrinkage problem becomes increasingly severe as the token set grows, leading to reduced representational diversity.

One possible reason about pretrained method underperforms the original approach at low token numbers is the gap between the discrete representations learned during pretraining and the continuous representations during finetuning, which poses challenges to the VQ learning process. However, this negative impact is outweighed by the benefits of our solution as the codebook size increases. Overall, our approach not only addresses token representation shrinkage but also unleashes the potential of VQ, further leveraging the benefits of a large codebook. Additionally, exploring how to mitigate the performance gap when the token number is low remains a worthy avenue for further investigation.

## 5.3 ImageNet-100 Results

**Tokenizer performance** Both types of original tokenizers exhibit a clear token representation shrinkage problem as shown in Tab. 1. For the tokenizer used in MaskGIT [4], we observe limited variation among tokens indicated by relatively small cosine distances (0.67 *vs.* 0.94). It reflects the high similarity between tokens. In addition, the tokenizer exhibits low perplexity (924.57 *vs.* 5311.88), suggesting that only a small subset of tokens is frequently utilized. Together, these observations

Table 1: **Performance evaluation of various tokenizers on the ImageNet-100 dataset.** "Shrink" indicates whether token representation shrinkage is present (✓) or mitigated using our proposed method (✗).

| Tokenizer | Shrink | r-FID ↓ | MSE ↓ | LPIPS ↓ | Cosine. ↑ | Perp. ↑ |
|-----------|--------|---------|-------|---------|-----------|---------|
| MaskGIT | ✗ | **8.58** | **3.28** | **2.34** | **0.94** | **5311.88** |
|  | ✓ | 12.22 | 3.91 | 2.70 | 0.67 | 924.57 |
| VAR | ✗ | **5.04** | **2.22** | **1.63** | **0.97** | **7044.51** |
|  | ✓ | 5.39 | 2.60 | 1.85 | 0.64 | 2801.88 |

Table 2: **ImageNet-100 generation**

| Model | Shrink | g-FID ↓ | Pixel Dist. ↑ |
|-------|--------|---------|---------------|
| MaskGIT | ✗ | **14.60** | **80.77** |
|  | ✓ | 14.75 | 75.89 |
| VAR | ✗ | **10.70** | **75.92** |
|  | ✓ | 12.88 | 70.69 |

Table 3: **ODIR generation**

| Model | Shrink | g-FID ↓ | Pixel Dist. ↑ |
|-------|--------|---------|---------------|
| VAR | ✗ | **34.33** | **49.83** |
|  | ✓ | 37.65 | 49.01 |

imply that token usage is poorly aligned with the embedding space, pointing to a clear case of token representation shrinkage. However, after pretraining, tokens are more evenly utilized and better aligned with the embedding space. These observations confirm that pretraining effectively mitigates the token representation shrinkage phenomenon. As a result, the pretrained tokenizer achieves improved reconstruction performance, with lower r-FID (8.58 *vs.* 12.22), LPIPS (2.34 *vs.* 2.70), and MSE (3.28 *vs.* 3.91).

A similar pattern is also observed for the multi-scale tokenizer in VAR. Without pretraining, severe token representation shrinkage is evident. Pretraining once again proves effective in alleviating this issue, leading to more balanced token usage and enhanced reconstruction performance.

**Generative Performance**    Token representation shrinkage significantly impairs the creativity of generative models, manifesting as a decline in both image quality and diversity as shown in Tab. 2. For the MaskGIT model, we observe that token representation shrinkage leads to a noticeable degradation in the generation FID (g-FID), indicating a reduction in the visual fidelity of synthesized images. Additionally, the pairwise pixel distance among generated samples is substantially reduced, suggesting that some outputs are highly similar. This phenomenon reflects a collapse in output variation, which we attribute directly to the narrowing of the token distribution(token representation shrinkage).

For the VAR model, we also observe a loss of creativity resulting from token representation shrinkage. Without proper mitigation, shrinkage in its multi-scale tokenizer leads to reduced generation quality and a clear drop in diversity. These results reinforce the conclusion that inadequate token representation limits the model's ability to capture the full generative distribution, ultimately compromising its overall creativity. The generated images are shown in Fig. 5.

## 5.4   Real-world Medical Data Results

To further validate our findings, we conduct experiments across different image modalities within the ODIR medical image dataset. For the VAR model, we again confirm the presence of token representation shrinkage as shown in the Tab. 4. Additionally, we observe a corresponding decline in generative performance, including noticeable reductions in both image quality and diversity(Table), consistent with our observations on natural image datasets.

However, for the MaskGIT model, the results deviate from our expectations. Despite clear evidence of token representation shrinkage in the tokenizer, the generated images do not exhibit a drop in creativity. This suggests a decoupling between token representation shrinkage and generation degradation in this particular setting. We hypothesize that this discrepancy may be attributed to the relatively small dataset size and limited inherent diversity within the ODIR dataset, which potentially

Table 4: **Performance evaluation of VAR tokenizer on the ODIR medical dataset.** "Shrink" indicates whether token representation shrinkage is present (✓) or mitigated using our proposed method (✗).

| Model | Shrink | r-FID ↓ | MSE ↓ | LPIPS ↓ | Cosine. ↑ | Perp. ↑ |
|-------|--------|---------|-------|---------|-----------|---------|
| VAR | ✗ | 11.04 | 2.05 | 6.79 | 0.90 | 5396.17 |
|     | ✓ | 10.91 | 2.57 | 8.79 | 0.62 | 940.55 |

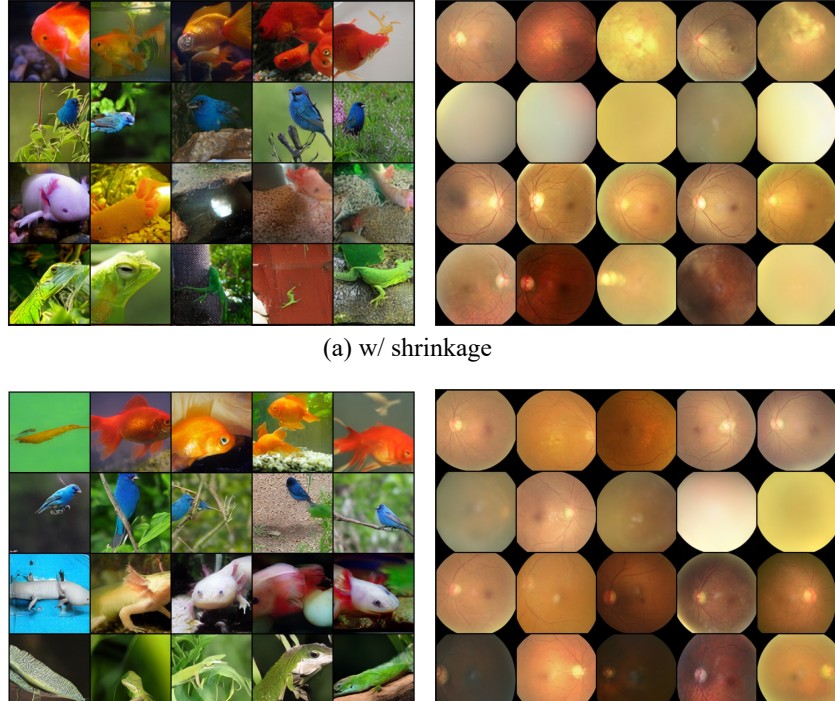

(a) w/ shrinkage

(b) w/o shrinkage

Figure 5: Generated images based on VAR. (a) ImageNet (a.left) and real-world medical images of eyes (a.right) generated using VAR as generative model and tokenizer **with token representation shrinkage**. (b) ImageNet (b.left) and real-world medical images of eyes (b.right) generated using VAR as generative model and tokenizer **without token representation shrinkage.**

masks the adverse effects of token representation. Detailed quantitative results are provided in the supplementary.

## 6 Conclusion

In this work, we systematically investigate the problem of token representation shrinkage in vector quantization, which is a critical yet overlooked factor contributing to mode collapse in transformer-based generative models. We demonstrate that commonly adopted token initialization strategies, especially those based on untrained encoders, lead to a collapse in token usage and embedding diversity, ultimately impairing the creativity of generative models by reducing output diversity and fidelity. To address this, we proposed a simple and effective two-stage training method that involves pretraining the encoder without VQ followed by fine-tuning with VQ. Our theoretical analysis and extensive experiments across synthetic, natural, and medical datasets confirm that this approach mitigates shrinkage, enhances token utilization, and improves generative performance. These findings highlight the importance of tokenizer design and initialization in discrete representation learning and open up new avenues for further research on improving generative expressiveness in VQ-based models.

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
