# OpenReview forum: "Token Representation Shrinkage Impairs Creativity of Generative Models"
_NeurIPS.cc/2025/Conference — Submitted to NeurIPS 2025_

### Official Review · Reviewer_DwYd · 2025-06-27

**Clarity:** 3
**Significance:** 1
**Originality:** 1
**Rating:** 4
**Confidence:** 3

**Summary:**

This paper addresses the issue of mode collapse in transformer-based generative models. The authors demonstrate that the tokenization strategy can diminish the creativity of these models, leading to mode collapse due to a phenomenon they term "token representation shrinkage." They suggest that this issue may partly arise from inadequate token initialization. To mitigate this, they propose a training process that initially excludes Vector Quantization (VQ) before fine-tuning with VQ. Their results show that this approach effectively reduces "token representation shrinkage" and enhances the overall diversity of the generative model.

**Questions:**

Could you please address the comments mentioned in the weaknesses section ?

**Ethical Concerns:**

["NO or VERY MINOR ethics concerns only"]

**Final Justification:**

The rebuttal has clarified my questions and addressed my concerns; therefore, I am raising my score to 4.

**Limitations:**

yes

**Quality:**

2

**Strengths And Weaknesses:**

Strength:
Mode collapse in transformer-based image generative models is a menaingful topic that is worth investigating time.

Weaknesses:
- The abstract doesn’t describe the full content presented in the paper since it doesn’t mention three out of the four contributions explained in line 59-66.
- In the Related work section, the authors talk about the issue of codebook collapse and seem to present it as a different issue than token representation shrinkage (in line 92-93). It is unclear to me why these two issues are different ones ?
- In Line 134, the authors mention (a), (c) and (e) in Fig. 3 but there is only (a) and (b). It is also unclear how the plot (b) w/o shrinkage was obtained.
- In the experimental part, the authors propose a strategy to improve vector quantization training and mitigate token representation shrinkage. However, there is no quantitative comparison to previous works also aiming at improving VQ training (such as the ones cited in related work in L86-94).
- The experimental setting focuses on relatively small dataset (CiFAR=10 and ImageNet-100) and it is not clear that the results would generalize well in a real-world scenario.
- To better measure image diversity and consistency, it would be interesting to also add recall and precision metrics

---

> ### Author Rebuttal · Authors · 2025-07-31
>
> # Reviewer 4
>
> We appreciate the constructive comments from reviewer **DwYd** . We provide responses to each individual question below.
>
> > W1: The abstract doesn’t describe the full content presented in the paper.
> >
>
> We have **updated the abstract to briefly summarize all four contributions** outlined in Lines 59–66.
>
> We like to emphasize that our main contribution in this work is **discovery of token representation shrinkage** phenomenon which was unknown to the community and is important for diversity of outputs of generative models.  In addition, **token representation shrinkage** is fundamentally different from codebook collapsing phenomenon (which we will clarify in the W2 section). We have also adjusted the abstract to emphasize this.
>
> > W2: What’s the difference of code collapse and token representation shrinkage
> >
>
> **These two phenomena are distinct issues in VQ and are fundamentally different.**  Codebook collapse refers to the deactivation of codes, while token representation shrinkage is the overall code distribution becomes narrower and fails to match the original embedding distribution regardless of code activation. **Codebook collapse**, as reported in many previous studies[1, 2, 3, 4], refers to the phenomenon where a portion of the codebook entries become "dead" (they are never or rarely used during training).  **In contrast, t**oken representation shrinkage **can occur even when all codebook entries remain active. Token representation shrinkage, which we first formally identified in our manuscript,** refers to a situation where the learned token embeddings crowd into a small region of the latent space, thereby limiting representational diversity.  Identifying the token representation shrinkage phenomenon is the major contribution in our study as it informs tokenizer design and behavior analysis of generative models.
>
> > W3: No (c) and (e) in Fig. 3.  It is also unclear how the plot (b) w/o shrinkage was obtained.
> >
>
> We sincerely apologize for the typographical error.  We have revised the figure reference to Fig. 3(a) accordingly in the updated manuscript.
>
> Regarding Fig. 3(b) ("w/o shrinkage"), this plot was obtained by initializing the codebook based on a trained encoder, as described in Lines 153–158. Since the pretrained encoder produces well-dispersed embeddings with semantic separability, it helps mitigate token representation shrinkage by promoting a more balanced token distribution early in training.
>
> > W4: No quantitative comparison to previous works also aiming at improving VQ training.
> >
>
> We appreciate the suggestion to include comparisons with prior work on improving VQ training
>
> We conducted additional experiments comparing the pretraining method with **SimVQ** [4](ICCV2025), targeting codebook collapse.
>
> As shown in Table 1, on CIFAR-10 with varying codebook sizes, our method achieves comparable reconstruction performance to SimVQ. And if we combine our method with SimVQ, it further improves the performance. That demonstrates that token representation shrinkage and code collapse are different problems. Addressing both of code collaspe and token representation will further improve the performance of vector quantization.
>
> Table 1. Ablation on CIFAR-10 across different codebook sizes.  MSE (×10^-4)
>
> |  | 4096 | 8192 | 16384 | 32768 | 65536 |
> | --- | --- | --- | --- | --- | --- |
> | Original | 10.153 | 9.653 | 9.716 | 9.716 | 9.693 |
> | Pretrained(Ours) | 10.124 | 9.227 | 8.456 | 8.001 | 7.54 |
> | Sim VQ | 10.237 | 9.305 | 8.548 | 7.961 | 7.412 |
> | Sim VQ + Pretrained | **9.693** | **8.817** | **8.1710** | **7.612** | **7.150** |
>
> On the ImageNet-100 dataset (Table 2), our method also demonstrates better reconstruction quality, highlighting the effectiveness of our strategy in relative large-scale scenarios.
>
> Due to resource limitation and time limitation, the generative results on Imagenet-100 have not finished yet.  The initial training  loss in the first few epochs are shown in Table. 3.
>
> Table 2. VAR tokenizer performance on Imagenet-100.
>
> |  | rFID | MSE(×10^-4) | LPIPS(×10^-3) |
> | --- | --- | --- | --- |
> | Original | 5.39 | 2.601 | 1.851 |
> | Pretrained(Ours) | 5.04 | 2.218 | 1.633 |
> | Sim VQ | 5.53 | 2.501 | 1.782 |
> | Sim VQ + Pretrained | **4.93** | **2.170** | **1.592** |
>
> Table 3. Training Loss(Cross Entropy) for VAR
>
> |  | epoch 2 | epoch 4 | epoch 6 | epoch 8 |
> | --- | --- | --- | --- | --- |
> | Sim VQ | 7.916 | 7.300 | 7.136 | 7.001 |
> | Sim VQ + Pretrained | 8.404 | 7.784 | 7.590 | 7.439 |
>
> Besides CIFAR-10 and Imagenet-100, we also conduct experiments on ODIR and Oxford Flower 102. The results as shown in Table 4 to Table 7. We notice that our simple solution improves the diversity of generative model on different dataset.  It is important to emphasize that the major contribution of this paper is the **discovery of the phenomenon of token representation** and understand **how it affect the diversity of generative models’ outputs**. We use a simplest method of pretraining encoder to show the effects. But the encoder pretraining method is not our major contribution.  Therefore, it won’t be surprising to see advanced method such as SimVQ show better enhancement of diversity.
>
> Table 4. VAR tokenizer performance on ODIR.
>
> | VAR | rFID | MSE(×10^-5) | LPIPS(×10^-4) |
> | --- | --- | --- | --- |
> | Original | 10.91 | 2.565 | 8.793 |
> | Pretrained(Ours) | 11.04 | 2.049 | 6.786 |
> | Sim VQ | **10.19** | 2.479 | 8.757 |
> | Sim VQ + Pretrained | 11.17 | **2.001** | **6.678** |
>
> Table 5. VAR generation results on ODIR
>
> | VAR | gFID | Pixel Dis | Recall | Precision |
> | --- | --- | --- | --- | --- |
> | Original | 37.65 | 49.01 | 0.1288 | 0.7968 |
> | Pretrained(Ours) | 34.32 | **49.83** | 0.1328 | **0.8464** |
> | Sim VQ | **32.98** | 44.34 | **0.1816** | 0.8216 |
> | Sim VQ + Pretrained | 35.62 | 48.12 | 0.1696 | 0.8168 |
>
> Table 6. VAR tokenizer performance on Oxford Flowers 102. *Cos Dis* represents Cosine distance in codebook. *Euc Dis* represents Euclidean distance of codebook.
>
> |   | rFID | MSE(×10^-4) | LPIPS(×10^-3) | Perplexity | Cos Dis. | Euc. Dis. |
> | --- | --- | --- | --- | --- | --- | --- |
> | Original | 22.69 | 3.479 | 3.151 | 2488 | 0.58 | 1.33 |
> | Pretrained(Ours) | 22.84 | 3.417 | 3.086 | **7131** | **0.99** | **6.50** |
> | Sim VQ | 22.17 | 3.300 | 3.071 | 5510 | 0.79 | 0.80 |
> | Sim VQ + Pretrained | **21.82** | **3.250** | **2.937** | 6012 | 0.80 | 1.03 |
>
> Table 7. VAR tokenizer performance on Oxford Flowers 102
>
> |  | gFID  | Pixel Dis | Recall  | Precision |
> | --- | --- | --- | --- | --- |
> | Original | 27.48 | **158.76** | 0.31875 | 0.749 |
> | Pretrained(Ours) | 27.84 | 158.16 | **0.34875** | 0.729 |
> | Sim VQ | 27.32 | 158.61 | 0.2825 | **0.754** |
> | Sim VQ + Pretrained | **27.17** | 156.31 | 0.3275 | 0.750 |
>
> > W5: The experimental setting focuses on relatively small dataset (CIFAR=10 and ImageNet-100)
> >
>
> We are currently conducting experiments on 256 × 256 resolution for Imagenet-1k. The preliminary results of the first few epoch are listed in Table 8 and 9. We come from an academic institution and only have access for 2-4 GPUs. Thus, we are still running the experiments can not obtain the final results in time.
>
> Table. 8 LPIPS(×10^-1) during training on Imagenet-1k
>
> |  | Epoch 2 | Epoch 4 | Epoch 6 | Epoch 8 |
> | --- | --- | --- | --- | --- |
> | Original | 3.931 | 2.682 | 2.366 | 2.225 |
> | Pretrained(Ours) | 2.357 | 2.134 | 2.025 | 1.975 |
>
> Table. 9 MSE(×10^-2) during training on Imagenet-1k
>
> |  | Epoch 2 | Epoch 4 | Epoch 6 | Epoch 8 |
> | --- | --- | --- | --- | --- |
> | Original | 3.968 | 3.435 | 3.202 | 3.082 |
> | Pretrained(Ours) | 3.070 | 2.898 | 2.806 | 2.752 |
>
> We sincerely thank the reviewer for highlighting this important direction. We plan to extend our study to ImageNet-1K and potentially other real-world scenarios and will include this  in the final version of the paper.
>
> > W6: add recall and precision metrics.
> >
>
> We appreciate the insightful feedback from the reviewer. We have included Precision and Recall scores as measurement in the main text and attached as in Table 10, Table 11 and Table 12. VQ with reduced token representation shrinkage shows higher recall in generate outputs in VAR for ODIR and MaskGIT for ImageNet dataset.
>
> Table. 10 VAR generative results on ODIR
>
> |  | g-FID | Recall | Precision |
> | --- | --- | --- | --- |
> | Pretrained(Ours) | **34.33** | **0.1328** | **0.8464** |
> | Original | 37.65 | 0.1288 | 0.7968 |
>
> Table. 11 MaskGIT generative results on Imagenet-100
>
> |  | g-FID | Recall | Precision |
> | --- | --- | --- | --- |
> | Pretrained(Ours) | **14.61** | **0.26** | 0.64 |
> | Original | 14.75 | 0.24 | **0.66** |
>
> Table 12 VAR generative results on Imagenet-100
>
> |  | g-FID | Recall | Precision |
> | --- | --- | --- | --- |
> | Pretrained(Ours) | **10.70** | 0.00 | **0.66** |
> | Original | 12.89 | 0.00 | 0.63 |
>
> ## References
>
> [1]  Yu, Jiahui, et al. "Vector-quantized image modeling with improved vqgan." *arXiv preprint arXiv:2110.04627* (2021).
>
> [2] Zheng, Chuanxia, and Andrea Vedaldi. "Online clustered codebook." *Proceedings of the IEEE/CVF International Conference on Computer Vision*. 2023.
>
> [3] Huh, Minyoung, et al. "Straightening out the straight-through estimator: Overcoming optimization challenges in vector quantized networks." *International Conference on Machine Learning*. PMLR, 2023.
>
> [4] Zhu, Yongxin, et al. "Addressing representation collapse in vector quantized models with one linear layer." *arXiv preprint arXiv:2411.02038* (2024).

---

> ### Author Response · Authors · 2025-08-04
>
> Dear Reviewer DwYd,
>
> We provide a summary of our responses for your convenience and sincerely appreciate your insightful feedback.
>
> **1. Difference between token representation shrinkage and code collapse:**
>
> - Token representation shrinkage: the distribution of code assignments becomes narrower and fails to match the distribution of the original encoder embeddings.
> - Code collapse: codes are rarely used or become inactive.
> - Regardless of whether code collapse occurs, token representation shrinkage always reduces the representational diversity.
>
> Our major contribution is the identification of previously unrecognized token representation shrinkage problem, along with a systematic analysis of its characteristics and impact.
>
> **2. Follow-up Experiments:**
>
> We conducted additional experiments according to your constructive suggestions:
>
> - Added SimVQ [1] (ICCV 2025) as a new baseline.
> - Include additional real-world datasets: ImageNet-1k, ODIR, and Oxford Flower 102.
> - We report both Recall and Precision metrics for comprehensive evaluation.
>
> In conclusion, our additional experiments (as shown in Rebuttal Table 1 - 12) further demonstrate that token representation shrinkage limits the representational diversity of codebook.
>
> [1] Zhu, Yongxin, et al. "Addressing representation collapse in vector quantized models with one linear layer." *arXiv preprint arXiv:2411.02038* (2024).

---

> > ### Comment · Reviewer_DwYd · 2025-08-07
> >
> > Thank you for your detailed answer. The rebuttal has clarified my questions and addressed my concerns; therefore, I am raising my score to 4.

---

> > > ### Author Response · Authors · 2025-08-07
> > >
> > > Thank you for your kind follow-up. We sincerely appreciate your recognition and the time you spent reviewing our work.

---

### Official Review · Reviewer_t9D3 · 2025-06-30

**Clarity:** 2
**Significance:** 3
**Originality:** 3
**Rating:** 3
**Confidence:** 4

**Summary:**

This paper finds that the "token representation shrinkage" phenomenon in trained vector quantization (VQ) models impairs the downstream generation diversity of VQ-based generative models such as autoregressive models and masked image generation models. The paper first conducts experiments on a toy synthetic dataset to validate the above hypothesis. It then proposes a two-stage method, "pretraining without VQ, fine-tuning with VQ", to solve the codebook collapse problem. Following experiments on the synthetic dataset and some real-world small-scale datasets verify that the proposed method can mitigate the codebook collapse problem.

**Questions:**

See weaknesses above.

---

Overall, the finding in this paper is insightful and novel to the field of VQ-based image generation, but its effectiveness in boosting the generation diversity of existing works is questionable under the current uncommon setup.

I can raise my score to borderline accept or clear accept if the above concerns (Weaknesses 1 and 3) are appropriately addressed during the discussion period.

**Ethical Concerns:**

["NO or VERY MINOR ethics concerns only"]

**Final Justification:**

After considering the rebuttal and discussions with authors, my final justification for the borderline reject rating is that:

The "Token Representation Shrinkage" finding is interesting and novel (to my understanding, it is different from the codebook collapse issue),

But whether this finding would appear or "Impair Creativity of Generative Models" in commonly adopted experimental setups is unclear (lacks directly comparable results to prior works).

**Limitations:**

Yes

**Paper Formatting Concerns:**

No paper formatting concerns

**Quality:**

2

**Strengths And Weaknesses:**

**Strengths**
1. The hypothesis that the untrained encoder leads to collapsed token representations in the codebook is insightful and novel to me.
2. It is satisfying to see a toy experiment conducted in Section 4.1 to validate the above hypothesis.
3. The proposed two-stage training solution is straightforward but makes sense to me and experiments demonstrate its effectiveness on solving the codebook-collapse problem.

**Weaknesses**
1. My major concern lies in the misaligned evaluation metrics and datasets with common image generation works such as MaskGIT [4] and VAR [25].
   - Evaluation metrics: Recall ("Improved precision and recall metric for assessing generative models". In
NeurIPS 2019) is a widely used metric to measure generation diversity, which is also used in [4] and [25]. The defined Cosine distance and Perplexity metrics in this paper only measure the representation overlap within the codebook, instead of the final generated images. And the defined "Pixel Dist." metric ignores real images and feature-level diversity (unlike Recall). It is recommended to report Recall numbers, which is widely used in existing literature. The other two missed evaluation metrics are IS and Precision (see [4] and [25]).
   - Datasets: [4] and [25] conduct experiments using ImageNet-1K with 256x256 and 512x512 resolution. It is acceptable to only conduct 256 resolution experiments if time is limited.

   I would like to see experiments that follow the setup of previous works and compare results with the numbers reported in previous papers (Table 1 of [4] and [25]). This is for checking whether the finding and method in this paper can help improve generation diversity under the common experimental setup.

2. A minor weakness is about the novelty of the proposed two-stage training solution.

   Recently, there exists some work (e.g. arxiv:2501.01423) leveraging existing pretrained vision models to constrain the latent space of the autoencoder during the autoencoder's training. Such methods, to some extent, weakened the novelty of the proposed "pretraining w/o VQ then fine-tuning w/ VQ" method in this paper.

3. Below are some comments on paper clarity (correct me if I'm wrong):
   - The illustration of Figure 1 can be clearer. What are the meanings of the vertical axes of the histograms? What do the different colors stand for?
   - The relationship between encoder representations (embeddings) and tokens in the codebook is not clear (Line 142-150 and Fig. 3). With untrained encoders, why are the embedding distributions in the bottom two figures of Fig. 3 (a) still well-expanded? And why are the token distributions (which are trained to be close to encoder embeddings) far from the embedding distributions in Fig. 3(a), unlike Fig. 3(b)?
   - The introduction of the common initialization strategy seems to be ambiguous (Line 116). To me, initialization stands for the weights initialization of the codebook before training, which is a uniform initialization in [common codebases](https://github.com/CompVis/taming-transformers/blob/3ba01b241669f5ade541ce990f7650a3b8f65318/taming/modules/vqvae/quantize.py#L230), not the "K-means" initialization. Only after reading the context about the proposed method could I guess that the "initialization" here may indicate the codebook status at the beginning of the VQ codebook training process.
   - Line 134: Fig.3 does not contain (c) (e).

---

> ### Author Rebuttal · Authors · 2025-07-31
>
> > W1: My major concern lies in the misaligned evaluation metrics and datasets with common image generation works such as MaskGIT [4] and VAR [25].
>
> > W1.1:  Evaluation metrics: Recall ("Improved precision and recall metric for assessing generative models". In NeurIPS 2019) is … [MASK]... It is recommended to report Recall numbers, which is widely used in existing literature. The other two missed evaluation metrics are IS and Precision (see [4] and [25]).
>
> We appreciate the insightful feedback from the reviewer.  We have included Recall, Precision and IS scores as measurement in the main text and attached as in Table 1, Table 2 and Table 3. VQ with reduced token representation shrinkage shows higher recall in generated outputs in VAR for ODIR and MaskGIT for ImageNet datasets.
>
>
> Table. 1 VAR generative results on ODIR
>
> |                  | g-FID     | IS   | Recall     | Precision  |
> | ---------------- | --------- | ---- | ---------- | ---------- |
> | Pretrained(Ours) | **34.33** | 1.84     | **0.1328** | **0.8464** |
> | Original         | 37.65     |  **2.01**    | 0.1288     | 0.7968     |
>
> Table. 3 MaskGIT generative results on Imagenet-100
>
> |                  | g-FID     | IS        | Recall   | Precision |
> | ---------------- | --------- | --------- | -------- | --------- |
> | Pretrained(Ours) | **14.61** | 48.09     | **0.26** | 0.64      |
> | Original         | 14.75     | **50.40** | 0.24     | **0.66**  |
>
> Table. 1 VAR generative results on Imagenet-100
> |                  | g-FID     | IS          | Recall | Precision |
> | ---------------- | --------- | ----------- | ------ | --------- |
> | Pretrained(Ours) | **10.70** | **52.60** | 0.00   | **0.66**  |
> | Original         | 12.89     | 47.43     | 0.00   | 0.63      |
>
> > W1.2: Datasets: MaskGIT and VAR conduct experiments using ImageNet-1K with 256x256 and 512x512 resolution. It is acceptable to only conduct 256 resolution experiments if time is limited.
>
> As shown in Table. 4 and Table. 5, we are currently conducting experiments on 256 × 256 resolution for Imagenet-1k. The preliminary results of the first few epoch are listed in table x and x. We come from an academic institution and only have access for 2-4 GPUs. Thus, we are still running the experiments can not obtain the final results in time.
>
> Table. 4 LPIPS during training on Imagenet-1k
>
> |                  | Epoch 2 | Epoch 4 | Epoch 6 | Epoch 8 |
> | ---------------- | ------- | ------- | ------- | ------- |
> | Original         | 3.931   | 2.682   | 2.366   | 2.225   |
> | Pretrained(Ours) | 2.357   | 2.134   | 2.025   | 1.975   |
>
> Table. 5 MSE during training on Imagenet-1k
>
> |                  | Epoch 2 | Epoch 4 | Epoch 6 | Epoch 8 |
> | ---------------- | ------- | ------- | ------- | ------- |
> | Original         | 3.968   | 3.435   | 3.202   | 3.082   |
> | Pretrained(Ours) | 3.070   | 2.898   | 2.806   | 2.752   |
>
> We sincerely thank the reviewer for highlighting this important direction. We plan to extend our study to ImageNet-1K and potentially other real-world scenarios and will include this in the final version of the paper.
>
> > W2: A minor weakness is about the novelty of the proposed two-stage training solution. Recently, there exists some work (e.g. arxiv:2501.01423) leveraging existing pretrained vision models to constrain the latent space of the autoencoder during the autoencoder's training. Such methods, to some extent, weakened the novelty of the proposed "pretraining w/o VQ then fine-tuning w/ VQ" method in this paper.
>
> We appreciate the reviewer’s observation and the pointer to related work. While similar two-stage strategies have emerged recently, we would like to emphasize that the **core contribution of our work lies not in the pretraining procedure itself**, but in the **identification and analysis of the token representation shrinkage phenomenon**. To the best of our knowledge, this issue has not been formally recognized in prior literature.
>
> > W3: Below are some comments on paper clarity (correct me if I'm wrong):
>
> > W3.1: The illustration of Figure 1 can be clearer. What are the meanings of the vertical axes of the histograms? What do the different colors stand for?
>
> Thank you for pointing out the lack of clarity in the figure caption. We have revised it to be more informative in the updated manuscript.. In **Figure 1**, the **vertical axis represents the density** of token distribution, i.e., how often each token appears across different image classes.
>
> The **cyan bars** indicate the ideal case where all classes are represented by a similar number of tokens, suggesting no token representation shrinkage.  The **red bars** indicate the presence of token representation shrinkage, where some classes are overrepresented by tokens while others receive disproportionately fewer tokens. We will add additional illustration to clearer explain the phenomena.
>
> > The relationship between encoder representations (embeddings) and tokens in the codebook is not clear (Line 142-150 and Fig. 3). With untrained encoders, why are the embedding distributions in the bottom two figures of Fig. 3 (a) still well-expanded? And why are the token distributions (which are trained to be close to encoder embeddings) far from the embedding distributions in Fig. 3(a), unlike Fig. 3(b)?
>
> The plots in **Fig. 3** actually show the **final states** **after training** the VQ-VAE model. The only difference between Fig. 3(a) and Fig. 3(b) is how the codebook was initialized:
>
> - In Fig. 3(a), the codebook was initialized from an **untrained encoder**.
> - In Fig. 3(b), the codebook was initialized from a **trained encoder** using K-means over its output.
>
> We observe that the encoder learns to produce semantically meaningful and well-dispersed embeddings after training, regardless of initialization. However, when the codebook is initialized from an **untrained encoder**, the initial codes are densely clustered. During training, the encoder updates rapidly, but the codebook updates lag behind and get trapped, leading to a situation where **only a** small subset of codebook entries tracks the encoder. As a result, the token becomes concentrated(**token representation shrinkage** occurs).
>
> > The introduction of the common initialization strategy seems to be ambiguous (Line 116). To me, initialization stands for the weights initialization of the codebook before training, which is a uniform initialization in [common codebases](https://github.com/CompVis/taming-transformers/blob/3ba01b241669f5ade541ce990f7650a3b8f65318/taming/modules/vqvae/quantize.py#L230), not the "K-means" initialization. Only after reading the context about the proposed method could I guess that the "initialization" here may indicate the codebook status at the beginning of the VQ codebook training process.
>
> We thank the reviewer for pointing out the ambiguity. We have renamed “codebook initialization” to “setting initial values of codebook embedding vectors” in the updated text to avoid possible confusion.
>
> > Line 134: Fig.3 does not contain (c) (e).
>
> We thank the reviewer for pointing out the typo and have revised the figure reference to Fig. 3(a) accordingly in the updated manuscript.
>
> ## References

---

> > ### Comment · Reviewer_t9D3 · 2025-08-01
> >
> > Thank the authors for the rebuttal, and I have read the rebuttal and the comments from other reviewers. My question regarding figure clarity has been addressed. But the concern about whether the claim and proposed method of this paper work in standard experimental settings still exists:
> >
> > 1. In Table "VAR generative results on Imagenet-100", why are the Recall numbers 0.00 for both the baseline and the proposed method?
> >
> > 2. Regarding experiments conducted under commonly adopted setups (e.g., ImageNet-1K 256x256), the current results are incomplete, i.e., there are no results directly comparable to the numbers reported in previous works' (MaskGIT or VAR) Tables.
> >
> > While I understand that the computational resources are limited and the discussion period is tight, the comparability with previous works is a point that should be considered before submission. I also note that this issue is a shared concern, as raised by Reviewer DwYd in the last two comments.

---

> > > ### Author Response · Authors · 2025-08-02
> > >
> > > Thank you for your valuable follow-up question and insightful observations.
> > >
> > > In the original table 'VAR generative results on ImageNet-100', we adopt the default evaluation setting to remain consistent with the protocols in \[1\]\[2\]. Specifically, we set the hyperparameter k=3 in k-nearest neighbors to construct the reference and sample manifolds for Recall computation. Under this default configuration (k=3), the Recall of original VAR and ours on ImageNet-100 are reported as 0.
> > >
> > > To further clarify this observation, we include results under **k = 3, 4, 5** in Table 1 below:
> > >
> > > Table 1. **Recall** under k=3,4,5.
> > >
> > > | **Model**        | **k = 3** | **k = 4** | **k = 5** |
> > > | ---------------- | --------- | --------- | --------- |
> > > | Pretrained(Ours) | 0.00      | 0.318     | 0.318     |
> > > | Original         | 0.00      | **0.345**     | **0.345**     |
> > >
> > > Table 2. **Coverage** under k=3,4,5.
> > >
> > > | **Model**        | **k = 3** | **k = 4** | **k = 5** |
> > > | ---------------- | --------- | --------- | --------- |
> > > | Pretrained(Ours) | **0.360**     | **0.448**     | **0.523**     |
> > > | Original         | 0.329     | 0.411     | 0.478     |
> > >
> > > From Table 1, we observe that the Recall metric of ours is lower than baseline VAR. However, as noted in [3], the theoretical basis for this setting remains unclear. Recent studies [3,4,5] have questioned the robustness and reliability of current Precision-Recall metrics in generative evaluation, particularly in the presence of outliers. This suggests that token representation shrinkage could cause the generation of distorted or incorrect images (outliers), which further leads to an overestimation of Recall. In contrast, Coverage\[3\] is less sensitive to such outliers and provides a more stable assessment.
> > >
> > > As suggested in \[3\], Coverage is considered a more reliable alternative to Recall and is therefore included in our evaluation. As shown in Table 2, with our pretraining solution, Coverage is consistently improved, indicating our solution can help the model learn a diverse and more representative distribution, and the token representation shrinkage issue is mitigated. We will include this analysis and discussion in the final manuscript to better clarify the limitations of current metrics and the observed evaluation behavior.
> > >
> > > [1] Tian, Keyu, et al. "Visual autoregressive modeling: Scalable image generation via next-scale prediction." *Advances in neural information processing systems* 37 (2024): 84839-84865.
> > >
> > > [2] Chang, Huiwen, et al. "Maskgit: Masked generative image transformer." *Proceedings of the IEEE/CVF conference on computer vision and pattern recognition*. 2022.
> > >
> > > [3] Naeem, Muhammad Ferjad, et al. "Reliable fidelity and diversity metrics for generative models." *International conference on machine learning*. PMLR, 2020.
> > >
> > > [4] Khayatkhoei, Mahyar, and Wael AbdAlmageed. "Emergent asymmetry of precision and recall for measuring fidelity and diversity of generative models in high dimensions." *International Conference on Machine Learning*. PMLR, 2023.
> > > [5] Liang, Yuanbang, et al. "Efficient precision and recall metrics for assessing generative models using hubness-aware sampling." *Proceedings of Machine Learning Research* 235 (2024): 29682-29699.

---

> > > > ### Comment · Reviewer_t9D3 · 2025-08-02
> > > >
> > > > I appreciate the follow-up explanation and acknowledge that what are better evaluation metrics for generative models remains an open question in the community.
> > > >
> > > > As mentioned in my previous round response, the comparability to prior works to check whether the claim and method work in standard settings is the reason that I cannot give higher scores. I decided to keep the borderline reject rating, and won't be upset if the paper is accepted.

---

> > > > > ### Author Response · Authors · 2025-08-05
> > > > >
> > > > > Thank you for your time and valuable feedback. We are pleased to have the opportunity to clarify the issues and address your concerns.

---

### Official Review · Reviewer_7kBV · 2025-07-01

**Clarity:** 4
**Significance:** 4
**Originality:** 4
**Rating:** 5
**Confidence:** 5

**Summary:**

This paper identifies token representation shrinkage -- the collapse of token diversity in VQ codebooks -- as a key factor impairing creativity (i.e., output diversity and quality) in transformer-based generative models. The authors attribute this to biased codebook initialization using untrained encoders. They propose a simple fix: pretrain the encoder without VQ, then fine-tune with VQ enabled. Experiments on synthetic data, CIFAR-10, ImageNet-100, and ODIR (medical images) show that this strategy improves token usage and generation quality across models like MaskGIT and VAR.

**Questions:**

1. This paper's focus on representation shrinkage during quantization is reminiscent of a related challenge in unsupervised hashing. In particular, the recent work by Ng et al. [1] on Similarity Distribution Calibration (SDC) addresses representation collapse in hash code space by regularizing the pairwise similarity distribution among hash codes to match a reference distribution. This reduces ambiguity in comparing hash codes and prevents over-concentration of representations.
Given the conceptual alignment between token representation shrinkage in VQ and hash code collapse in binary embeddings, it would be insightful to discuss whether such distribution-shaping techniques (e.g., pairwise similarity regularization or target entropy matching) could complement or further improve your proposed method. A comparative analysis or at least a short discussion could enhance the paper’s positioning and show awareness of related mitigation strategies outside generative modeling.
2. do you also resize CIFAR10 dataset to 128x128? or perform with 32x32? if 32x32, what is latent size?
3. what happens if the pretrained encoder is undertrained before VQ is introduced? Understanding the sensitivity to pretraining quality could clarify how much training is "enough" before enabling VQ.
4. feel free to look at weaknesses above if it is appropriate to answer.

[1] Ng et al. Unsupervised Hashing with Similarity Distribution Calibration. BMVC 2023.

**Ethical Concerns:**

["NO or VERY MINOR ethics concerns only"]

**Final Justification:**

1. good insight paper -- a "preprocessing" to VQ space is needed to avoid code collapsing
2. good discussion with [1] -- both works are doing similar direction, although one is on hashing and one is on image VQ. this implies that there are still a lot of areas to be improved in image tokenizer design.

- final question but not related my justification: would be interesting to apply such ditsribution level regularization [1] in learning the VQ codebook towards a end-to-end design

i am raising my score to accept.

**Limitations:**

yes. the author mentioned that the proposed solutio effectively mitigates token representation shrinkage in most settings, but suboptimal when the codebook size is small. They provided a preliminary analysis of this behavior and suggest that addressing this limitation presents an interesting direction for future research.

**Quality:**

3

**Strengths And Weaknesses:**

Quality:
- The authors conduct a wide range of solid experiments across synthetic, natural (CIFAR-10, ImageNet-100), and medical (ODIR) datasets.
- The paper includes clean comparisons between models trained with and without the proposed pretraining strategy, showing consistent improvements across metrics such as perplexity, cosine similarity, reconstruction FID, and generation FID.
- A clear mathematical formulation links token representation shrinkage to decreased entropy and increased quantization error.

Clarity:
- The paper is well-organized and includes intuitive diagrams (e.g., Figures 1–3) that illustrate the shrinkage problem and its consequences. The motivation, methodology, and experimental design are easy to follow.

Significance:
- Highlights a critical bottleneck in generative modeling -- by identifying a fundamental failure mode in the tokenizer itself, this paper exposes an overlooked source of mode collapse. This finding could guide improvements in generative model design, especially for VQ-based systems.
- Applicable to multiple models and domains: Demonstrated benefits across MaskGIT and VAR architectures, and across domains (natural and medical images), indicate a wide scope of impact.
- The proposed fix is lightweight and easy to implement in existing pipelines, making it likely to influence real-world systems where generation diversity is critical (e.g., art, design, healthcare).

Originality:
- New framing of mode collapse. While prior work has largely focused on architectural or training objective improvements, this paper identifies an overlooked root cause -- biased initialization in tokenizers -- introducing the new term token representation shrinkage.
- Simple but novel solution. The idea of encoder pretraining prior to VQ is conceptually straightforward but has not been widely explored or formalized in this context. This training steps yields significant benefits without modifying model architecture.

Weaknesses:
- Simplistic creativity metrics: Creativity is evaluated using low-level measures like FID, LPIPS, and pairwise pixel distances. These do not fully capture semantic or perceptual novelty, especially in creative applications.
- No exploration of complementary strategies: The focus is entirely on initialization. Alternative solutions (e.g., entropy regularization since you analyzed the entropy) are not discussed or tested. A worth alternative way is to apply/modify the loss proposed in SDC [1] (see more at Questions)
- The visual examples are limited. Including more qualitative outputs from models exhibiting shrinkage would help support the argument for reduced generative diversity. Fig. 5 at the moment is not enough to illustrate the shrinkage problem -- how is that creative/not creative?

[1] Ng et al. Unsupervised Hashing with Similarity Distribution Calibration. BMVC 2023.

---

> ### Author Rebuttal · Authors · 2025-07-31
>
> # Reviewer 2
>
> > Q1:
> >
>
> We sincerely thank the reviewer for this insightful observation and the pointer to the **Similarity Distribution Calibration (SDC)** method by Ng et al. (BMVC 2023). Indeed, both **token representation shrinkage in vector quantization** and **similarity collapse in hashing** stem from **over-concentration of representations in a limited space**, leading to degraded semantic separability.
>
> SDC addresses this issue by aligning the pairwise similarity distribution of binary codes with a predefined reference distribution (e.g., Beta distribution), thereby **reshaping the similarity structure globally** rather than preserving individual pairwise distances. This approach aligns well with our motivation to **encourage token diversity** and prevent overuse of a small subset of tokens.
>
> While our method currently mitigates shrinkage via pretrained strategies, **distribution-level regularization techniques like SDC offer a promising complementary direction**. Specifically, incorporating pairwise similarity distribution regularization into the VQ training process could potentially enhance token dispersion and further stabilize training.
>
> We will add a **discussion paragraph in the final version of the paper** to highlight this conceptual alignment and outline it as an exciting direction for future exploration.
>
> > Q2: do you also resize CIFAR10 dataset to 128x128? or perform with 32x32? if 32x32, what is latent size?
> >
>
> The image size is 32 * 32. And the latent size is 8 * 8.
>
> > Q3: what happens if the pretrained encoder is undertrained before VQ is introduced? Understanding the sensitivity to pretraining quality could clarify how much training is "enough" before enabling VQ.
> >
>
> Table 1. Validation loss(MSE ×10^-3) for AE.
>
> | Epoch | 0 | 10 | 20 | 40 | 60 | 80 | 100 |
> | --- | --- | --- | --- | --- | --- | --- | --- |
> | Val loss | 2.44 | 4.41 | 2.63 | 1.43 | 1.09 | 0.76 | 0.63 |
>
> Table 2.  Performance of VQVAE initialized by different AE.
>
> | AE Epoch | 0 | 10 | 20 | 40 | 60 | 80 | 100 |
> | --- | --- | --- | --- | --- | --- | --- | --- |
> | MSE (×10^-3) | 9.65 | 9.32 | 9.23 | 9.30 | 9.23 | 9.20 | 9.38 |
> | Perplexity | 2018.63 | 3127.93 | 3143.12 | 3126.99 | 3089.81 | 3107.58 | 3039.83 |
>
> Thank you for the insightful question. We conducted a simple ablation to study the effect of pretraining quality. Our results show that **even a short pretraining phase (e.g., 10 epochs)** is sufficient to significantly improve the performance of VQ-VAE. Beyond **20 epochs**, we observe that continued training of the autoencoder yields only marginal gains in VQ-VAE performance.
>
> This suggests that the benefits of pretraining plateau after a certain point, and **moderate pretraining is already sufficient** to mitigate token representation shrinkage.

---

> > ### Author Response · Authors · 2025-08-05
> >
> > Dear Reviewer **7kBV**
> >
> > Thanks again for your valuable feedback. And we provide a summary of our responses below for your convenience:
> > 1. Similarity collapse[1] and token representation shrinkage reflect analogous underlying phenomena. We have added a discussion paragraph to highlight the Similarity Distribution Calibration method as a promising future direction for addressing the issue of token representation shrinkage.
> > 2. The undertrained AE will also yield benefits to mitigate token representation shrinkage based on our additional experiments.
> >
> > We sincerely appreciate your acknowledgement of this work and look forward to your constructive feedback.

---

### Official Review · Reviewer_Zc2K · 2025-07-03

**Clarity:** 3
**Significance:** 4
**Originality:** 3
**Rating:** 5
**Confidence:** 4

**Summary:**

The authors identify the issue of token representation shrinkage when it comes to transformer-based generative models, which reduces their creativity when producing diverse and novel outputs. This issue is mainly attributed to biased initialization methods. They then provide a strategy based on vector quantization, which converts continuous representations into discrete tokens, to solve the problem by first pretraining without vector quantization and then finetuning with vector quantization.

**Questions:**

1. Could the authors explain the difference between codebook collapse and token representation shrinkage? If they are the same or similar things, are there other baseline methods to deal with codebook collapse that the authors could potentially compare against?
2. Given the claim that "relatively small dataset size and limited inherent diversity" masks the impact of shrinkage, can the authors provide empirical support for this hypothesis or quantify dataset size/diversity in any way to better understand the impact?
3. Can the authors discuss why r-FID goes up with the VAR model on the ODIR dataset when shrinkage is not present?
4. If the initialization strategy has an effect on token representation shrinkage, why did the authors not try different initialization methods to see if the problem can be alleviated?

Addressing these questions effectively may help raise my score.

**Ethical Concerns:**

["NO or VERY MINOR ethics concerns only"]

**Final Justification:**

The authors have addressed my questions effectively and I will therefore be raising my score from a 4 to a 5.

**Limitations:**

Yes.

**Paper Formatting Concerns:**

- Explain what the up and down arrows in the tables mean.
- Line 290: typo -> (Table)
- No descriptions in Tables 2 and 3.
- No bolding in Table 4.
- More uniform usage of Token vs. Model in Tables 1-4.

**Quality:**

3

**Strengths And Weaknesses:**

Strengths:
- The paper is clear and very well-written.
- The authors identify token representation shrinkage, a previously underexplored issue, as a phenomenon that limits the creativity of generative models.
- The formal definition of token representation shrinkage is intuitive and connects to the ideas of entropy and quantization error, which affect generation quality.
- The proposed solution is simple and intuitive.
- Good validation of token representation shrinkage in synthetic experiments and CIFAR-10 experiments. Good prevention of token representation shrinkage and good generative performance in ImageNet-100 with both VAR and MaskGIT.
- Good choice of evaluation metrics used to compare against baseline methods.

Weaknesses:
- It would be good to have a section on the specific solution to the token representation shrinkage problem - the authors allude to the two-stage training process and discuss its effectiveness, but it is not explicitly detailed anywhere.
- Although the authors report good generative performance with the ODIR dataset using VAR, they report that r-FID increases with token representation shrinkage, which does not line up with the authors' hypothesis. They do not mention why this is.
- The authors also mention that the results using MaskGIT are poor and do not report them at all; it may be worth reporting those as well.
- More of an understanding is required on why "relatively small dataset size and limited inherent diversity... [mask] the adverse effects of token representation." Especially since this is almost always the case in real-world medical datasets.
- More details on the difference between token representation shrinkage and codebook collapse could be useful.
- Tokenizer performance improvements seem to generally be greater than generation performance, which is arguably more important for generative models.

---

> ### Author Rebuttal · Authors · 2025-07-31
>
> # Reviewer 1
>
> > Q1: Could the authors explain the difference between codebook collapse and token representation shrinkage? If they are the same or similar things, are there other baseline methods to deal with codebook collapse that the authors could potentially compare against?
>
> First, the codebook collapse [1, 2, 3, 4] and token representation shrinkage are **distinct problems**. **Codebook collapse** refers to the phenomenon where a portion of the codebook entries become "dead" (they are never or rarely used during training). **In contrast, token representation shrinkage, which we newly formally identified in our manuscript,** refers to a situation where the learned token embeddings crowd into a small region of the latent space, thereby limiting representational diversity. Token representation shrinkage **can occur even when all codebook entries remain active.** In such cases, although the codebook appears active, the effective token usage is still limited, resulting in a loss of expressiveness. Identifying the token representation shrinkage phenomenon is the major contribution in our study as it informs tokenizer design and behavior analysis of generative models.
>
> In addition to clarifying that token representation shrinkage and code collapse are distinct issues, we further provide comparison results with SimVQ [4].
>
> Table 1. Ablation on CIFAR-10 across different codebook sizes.  MSE (×10^-4)
>
> |                     | 4096      | 8192      | 16384      | 32768     | 65536     |
> | ------------------- | --------- | --------- | ---------- | --------- | --------- |
> | Original            | 10.153    | 9.653     | 9.716      | 9.716     | 9.693     |
> | Pretrained(Ours)    | 10.124    | 9.227     | 8.456      | 8.001     | 7.54      |
> | Sim VQ              | 10.237    | 9.305     | 8.548      | 7.961     | 7.412     |
> | Sim VQ + Pretrained | **9.693** | **8.817** | **8.1710** | **7.612** | **7.150** |
>
> On the ImageNet-100 dataset (Table 2), our method also demonstrates better reconstruction quality, highlighting the effectiveness of our strategy in relative large-scale scenarios.
>
> Due to resource limitation and time limitation, the generative results on Imagenet-100 have not finished yet.  The initial validation loss in the first few epochs are shown in Table. 3.
>
> Table 2. VAR tokenizer performance on Imagenet-100.
>
> |                     | rFID     | MSE(×10^-4) | LPIPS(×10^-3) |
> | ------------------- | -------- | ----------- | ------------- |
> | Original            | 5.39     | 2.601       | 1.851         |
> | Pretrained(Ours)    | 5.04     | 2.218       | 1.633         |
> | Sim VQ              | 5.53     | 2.501       | 1.782         |
> | Sim VQ + Pretrained | **4.93** | **2.170**   | **1.592**     |
>
> Table 3. Training Loss(Cross Entropy) for VAR
>
> |                     | epoch 2 | epoch 4 | epoch 6 | epoch 8 |
> | ------------------- | ------- | ------- | ------- | ------- |
> | Sim VQ              | 7.916   | 7.300   | 7.136   | 7.001   |
> | Sim VQ + Pretrained | 8.404   | 7.784   | 7.590   | 7.439   |
>
> Besides CIFAR-10 and Imagenet-100, we also conduct experiments on ODIR and Oxford Flower 102. The results as shown in Table 4 to Table 7. We notice that our simple solution improves the diversity of generative model on different dataset.  It is important to emphasize that the major contribution of this paper is the **discovery of the phenomenon of token representation** and understand **how it affect the diversity of generative models’ outputs**. We use a simplest method of pretraining encoder to show the effects. But the encoder pretraining method is not our major contribution.  Therefore, it won’t be surprising to see advanced method such as SimVQ show better enhancement of diversity.
>
> Table 4. VAR tokenizer performance on ODIR.
>
> | VAR                 | rFID      | MSE(×10^-5) | LPIPS(×10^-4) |
> | ------------------- | --------- | ----------- | ------------- |
> | Original            | 10.91     | 2.565       | 8.793         |
> | Pretrained(Ours)    | 11.04     | 2.049       | 6.786         |
> | Sim VQ              | **10.19** | 2.479       | 8.757         |
> | Sim VQ + Pretrained | 11.17     | **2.001**   | **6.678**     |
>
> Table 5. VAR generation results on ODIR
>
> | VAR                 | gFID      | Pixel Dis | Recall     | Precision  |
> | ------------------- | --------- | --------- | ---------- | ---------- |
> | Original            | 37.65     | 49.01     | 0.1288     | 0.7968     |
> | Pretrained(Ours)    | 34.32     | **49.83** | 0.1328     | **0.8464** |
> | Sim VQ              | **32.98** | 44.34     | **0.1816** | 0.8216     |
> | Sim VQ + Pretrained | 35.62     | 48.12     | 0.1696     | 0.8168     |
>
> Table 6. VAR tokenizer performance on Oxford Flowers 102. *Cos Dis* represents Cosine distance in codebook. *Euc Dis* represents Euclidean distance of codebook.
>
> |                     | rFID      | MSE(×10^-4) | LPIPS(×10^-3) | Perplexity | Cos Dis. | Euc. Dis. |
> | ------------------- | --------- | ----------- | ------------- | ---------- | -------- | --------- |
> | Original            | 22.69     | 3.479       | 3.151         | 2488       | 0.58     | 1.33      |
> | Pretrained(Ours)    | 22.84     | 3.417       | 3.086         | **7131**   | **0.99** | **6.50**  |
> | Sim VQ              | 22.17     | 3.300       | 3.071         | 5510       | 0.79     | 0.80      |
> | Sim VQ + Pretrained | **21.82** | **3.250**   | **2.937**     | 6012       | 0.80     | 1.03      |
>
> Table 7. VAR tokenizer performance on Oxford Flowers 102
>
> |                     | gFID      | Pixel Dis  | Recall      | Precision |
> | ------------------- | --------- | ---------- | ----------- | --------- |
> | Original            | 27.48     | **158.76** | 0.31875     | 0.749     |
> | Pretrained(Ours)    | 27.84     | 158.16     | **0.34875** | 0.729     |
> | Sim VQ              | 27.32     | 158.61     | 0.2825      | **0.754** |
> | Sim VQ + Pretrained | **27.17** | 156.31     | 0.3275      | 0.750     |
>
> > Q2: Given the claim that "relatively small dataset size and limited inherent diversity" masks the impact of shrinkage, can the authors provide empirical support for this hypothesis or quantify dataset size/diversity in any way to better understand the impact?
>
> Table 8. Mean of cosine distance for visual features on Imagenet-100 and ODIR
>
> |          | ODIR | Imagenet-100 |
> | -------- | ---- | ------------ |
> | Cos. dis | 0.23 | 0.58         |
>
> Table 9. Shrinkage ratio for ODIR and Imagenet-100
>
> |                 | ODIR   | Imagenet-100 |
> | --------------- | ------ | ------------ |
> | Shrinkage Ratio | 0.9835 | 0.8465       |
>
> ODIR is ****a medical dataset containing only **retinal images**, which are naturally more homogeneous in content. To quantify **dataset diversity**, we computed the **mean cosine distance** between Inception-V3 features of test samples from both datasets. As shown in Table 8, ODIR exhibits significantly lower intra-dataset feature distances than ImageNet-100, indicating much lower visual diversity.
>
> Furthermore, to assess the impact of **token representation shrinkage** ratio on output diversity, we computed the ratio of **mean pixel distance between generated images with and without shrinkage** As shown in Table 9, the shrinkage ratio in pixel-level diversity due to shrinkage is substantially more pronounced on ImageNet-100 than on ODIR.
>
> > Q3: Can the authors discuss why r-FID goes up with the VAR model on the ODIR dataset when shrinkage is not present?
>
> We thank the reviewer for this thoughtful question. We offer the following possible explanations for the observed r-FID goes up. First, the ODIR dataset is relatively small(only 1200 images for test dataset), which may introduce fluctuations in FID measurements. Notably, as shown in the Table. 4  other metrics such as **MSE and LPIPS, consistently improve** after applying our pretraining strategy, indicating better reconstruction quality. Another reason is that the FID metric relies on features extracted from an Inception-V3 network pretrained on **ImageNet-1K**, which may not capture semantic differences accurately in **domain-specific medical images** such as fundus photos. This may affect the reliability of both FID and r-FID in this setting.
>
> > Q4: If the initialization strategy has an effect on token representation shrinkage, why did the authors not try different initialization methods to see if the problem can be alleviated?
>
> We thank the reviewer for the valuable suggestion. We would like to clarify that the **primary contribution of this work is the identification and analysis of the token representation shrinkage phenomenon**, which has not been formally studied before.
>
> To support this finding, we propose a **simple yet effective pretraining-based initialization strategy** that not only demonstrates the existence of shrinkage but also partially mitigates it. Our intention is not to exhaustively benchmark all possible initialization schemes, but rather to provide an initial and interpretable solution that highlights the significance of the problem.
>
> We believe that exploring alternative initialization strategies is a meaningful next step, and we plan to investigate and compare them in future work.
>
> ## References
>
> [1]  Yu, Jiahui, et al. "Vector-quantized image modeling with improved vqgan." *arXiv preprint arXiv:2110.04627* (2021).
>
> [2] Zheng, Chuanxia, and Andrea Vedaldi. "Online clustered codebook." *Proceedings of the IEEE/CVF International Conference on Computer Vision*. 2023.
>
> [3] Huh, Minyoung, et al. "Straightening out the straight-through estimator: Overcoming optimization challenges in vector quantized networks." *International Conference on Machine Learning*. PMLR, 2023.
>
> [4] Zhu, Yongxin, et al. "Addressing representation collapse in vector quantized models with one linear layer." *arXiv preprint arXiv:2411.02038* (2024).

---

> ### Author Response · Authors · 2025-08-06
>
> Dear Reviewer **Zc2K**
>
> Thank you for your thoughtful and constructive feedback. We truly appreciate the time and effort you dedicated to reviewing our work.
>
> You kindly suggested that addressing a particular concern might help improve the paper's evaluation. We have revised the manuscript accordingly and would be grateful if you could let us know whether our response resolves the issue.
>
> If you have any remaining questions or concerns, we are pleased to provide further clarification. Thank you again for your time and valuable feedback.

---

> > ### Comment · Reviewer_Zc2K · 2025-08-07
> > **Reply to rebuttal**
> >
> > The authors have addressed my questions effectively and I will therefore be raising my score from a 4 to a 5.

---

> > > ### Author Response · Authors · 2025-08-07
> > >
> > > Thank you again for your kind response. We sincerely appreciate your thoughtful feedback and the time you spent reviewing our work.

---

### Note · Authors · 2025-08-12

We sincerely appreciate the constructive feedback provided during the review and discussion, and thank all reviewers as well as AC for their time and effort.


In summary, our major contributions are:

a) identifying the previously unrecognized token representation shrinkage phenomenon

b) providing a systematic analysis of its characteristics and impact

c) proposing a simple and effective pretraining strategy to mitigate shrinkage.


Our responses are summarized as follows:

1. We clarified the conceptual difference and improved the presentation

- Clarified the distinction between *token representation shrinkage* and *codebook collapse*. The former refers to a mismatch between codes and embeddings, while the latter describes codes that are inactive or rarely used.
- Revised the manuscript to improve figure clarity, elaborate on key technical points, and correct minor typographical and formatting errors.

2. We conducted further experiments and analyses, including:

- **Pretraining strategy v.s. SimVQ[1]:**
    - Results show that combining the two methods yields better performance than employing either method individually.
    - It suggests that token representation shrinkage is distinct from codebook collapse.
- **Additional metric evaluation:** Besides FID and pixel distance, we incorporated Recall, Precision, and IS for both VAR and MaskGIT, further showing that shrinkage impairs generative model creativity.
- **Validation of shrinkage across diverse datasets:** Conducted experiments on CIFAR-10, ODIR, Oxford Flowers 102, ImageNet-100, and ImageNet-1K, showing that shrinkage degrades tokenizer performance and the generative model’s creativity. For ImageNet-1K, preliminary results exhibit similar trends, though full experiments were not completed due to limited resources and time.
- **Role of pretraining in shrinkage:** Verified that even an undertrained encoder can help mitigate token representation shrinkage.

[1] Zhu, Yongxin, et al. "Addressing representation collapse in vector quantized models with one linear layer." *arXiv preprint arXiv:2411.02038* (2024).

---

### Decision · Program_Chairs · 2025-09-17

**Decision:**

Reject

**Comment:**

The paper introduces and analyzes the concept of “token representation shrinkage” in VQ-based generative models, claiming that it limits creativity by reducing token diversity in codebooks. The authors demonstrate the phenomenon across synthetic, natural, and medical datasets, and propose a simple two-stage training method (encoder pretraining before VQ) as a mitigation. Reviewers acknowledged that the contribution is conceptually novel and the paper is generally well written, with clear motivation, intuitive formulations, and experiments showing consistent improvements on small- to medium-scale datasets.

However, significant weaknesses remain. Multiple reviewers noted that the evaluation setup is misaligned with standard benchmarks: key experiments on ImageNet-1K at higher resolutions are incomplete, and comparisons with prior state-of-the-art results are missing. The creativity assessment relies heavily on FID, LPIPS, and pixel-level distances, without robust semantic or perceptual diversity measures. While the authors added Recall, Precision, and IS during rebuttal, results are either inconclusive (e.g., recall scores reported as zero) or incomplete, leaving doubts about whether the claimed phenomenon impairs creativity in widely adopted setups. Questions of novelty were also raised, since similar two-stage initialization strategies already exist, and the proposed fix may be viewed as incremental. Overall, the rebuttal provided clarifications and some additional experiments, but the core concerns about comparability, robustness of evaluation, and incomplete large-scale validation were not fully resolved.

In conclusion, while the idea of token representation shrinkage is intriguing and potentially impactful, the submission remains borderline due to limited evidence that the finding holds under standard experimental conditions and the lack of strong baselines. The weaknesses outweigh the strengths at this stage, and the work does not yet appear ready for NeurIPS. A more complete version, with thorough large-scale validation and stronger evaluation against existing work, would be better suited for submission to another venue. Therefore, the AC recommends rejection.